# The relationships between impulsivity and mood in bipolar disorder: An ecological momentary assessment study

Almudena Ramírez-Martín[1,2,3], Lea Sirignano[4], Jerome C. Foo[4,5,6], Fabian Streit[4,7], Josef Frank[4], Stephanie H. Witt[4], Marcella Rietschel[4], Fermin Mayoral-Cleries[1,2], Berta Moreno-Küstner[2,3], Jose Guzmán-Parra[1,2]*

1 Department of Mental Health, University General Hospital of Malaga, Malaga, Spain, 2 Biomedical Research Institute of Malaga (IBIMA-PlataformaBionand), Malaga, Spain, 3 Department of Personality, Assessment and Psychological Treatment, University of Málaga, Andalusian Group of Psychosocial Research (CTS-945), Málaga, Spain, 4 Department of Genetic Epidemiology in Psychiatry, Medical Faculty Mannheim, Central Institute of Mental Health, University of Heidelberg, Mannheim, Germany, 5 Department of Psychiatry, College of Health Sciences, University of Alberta, Edmonton, Canada, 6 Neuroscience and Mental Health Institute, University of Alberta, Edmonton, Canada, 7 Hector Institute for Artificial Intelligence in Psychiatry, Central Institute of Mental Health, Medical Faculty Mannheim, Heidelberg University, Mannheim, Germany

* jose.guzman.parra.sspa@juntadeandalucia.es

## Abstract

### Background

Impulsivity is a key feature of bipolar disorder (BD) associated with various negative outcomes. Recent use of ecological momentary assessment (EMA) has allowed for nuanced examination of the mechanisms of mood and impulsivity dysregulation. However, few existing studies have used an ecological momentary assessment of impulsivity in multiplex families with BD and examined its associations with mood.

### Objective

Using EMA, this study investigated the concurrent and predictive relationships between impulsivity and mood.

### Methods

Multiplex family members with BD (BDF, n = 8), unaffected family members (FC, n = 6), individuals with BD not from families (BDC, n = 8) and healthy controls (HC, n = 8), completed daily EMA surveys about mood and impulsivity for 6–12 weeks. Mixed-effects regression concurrent and lagged models were employed to analyze the relationship between impulsivity and mood.

provided the original author and source are credited.

**Data availability statement:** The datasets are available at the following link: https://osf.io/6qsbx/

**Funding:** The study was supported by the German Research Foundation (DFG) and the German Federal Ministry of Education and Research (BMBF) through the Integrated Network IntegraMent under the auspices of the e:Med programme [grants 01ZX1314A to MMN and SC; 01ZX1314G and 01ZX1614G to MR], through "ASD-Net" [grant 01EE1409C to MR and SHW]; by ERA-NET NEURON through "SynSchiz - Linking synaptic dysfunction to disease mechanisms in schizophrenia – a multilevel investigation" [grant 01EW1810 to MR] and "EMBED - impact of Early life MetaBolic and psychosocial strEss on susceptibility to mental Disorders; from converging epigenetic signatures to novel targets for therapeutic intervention" [grant 01EW1904]; "FOR2107" [grants RI908/11-2 to MR; NO246/10-2 to MMN and WI3439/3-2 to SHW]; by the Andalusian regional Health and Innovation Government [grants PI-0060-2017]. MMN is a member of the DFG-funded cluster of excellence ImmunoSensation. The funders had no role in study design, data collection and analysis, decision to publish, or preparation of the manuscript.

**Competing interests:** The authors have declared that no competing interests exist.

**Abbreviations:** AbiF, Andalusian Bipolar Family study; ANOVA, analysis of variance; BD, bipolar disorder; BDC, bipolar disorder ambulatory cases; BDF, bipolar disorder participants from families; EMA, ecological momentary assessment; FC, family controls; HC, healthy controls.

## Results

The BDF (Diff = −31.70, $p = 0.001$) and BDC (Diff = −25.74, $p = 0.007$) groups had a significantly lower mean in mood scores compared to the HC group but not compared to the FC group. There were no significant differences in the mean impulsivity scores between the groups. Time-lagged analyses revealed a significant negative association between prior impulsivity and mood at the next assessment independent of diagnosis (OR=0.939, $p = 0.002$). However, the opposite relationship between prior mood and impulsivity was not significant (OR=0.996, $p = 0.135$).

## Conclusions

These results contribute to the understanding of the complex interactions between BD, the genetic load of the disorder, impulsivity and mood. Furthermore, these findings indicate the potential benefits of addressing impulsivity as a means to improve mood outcomes at an early stage.

## Introduction

Impulsivity is a frequently presenting component of bipolar disorder (BD) in its different phases and episodes, and has been proposed as a core feature of the disorder [1,2]. Furthermore, given the strong genetic load of BD, impairments of impulsivity have been observed in unaffected first-degree relatives [3,4]. Considering the important role of impulsivity in BD and its association with non adherence to medication, lower quality of life, higher functional disability, longer duration of illness and an increased number of suicide attempts [5–9], it is relevant to understand the relationship of impulsivity with different mood states.

Impulsivity is a complex and multidimensional construct without a widely agreed uponstructure [10]. Several researchers have attempted to categorize it using different paradigms and have distinguished between: a) trait impulsivity and state impulsivity [11]; b) behavioural impulsivity versus self-reported impulsivity [10]; and c) impulsive choice and impulsive action [12]. Generally, impulsivity can be defined as a predisposition to rapid, unplanned reactions to internal or external stimuli that fail to take into account the negative consequences of those reactions to the individual themself or to others [13]. Its various dimensions have been commonly assessed with both self-report measures (e.g., the Barratt Impulsiveness Scale) [14] and behavioural measures (behavioural laboratory tasks). However, despite the enormous potential of these methods,none of them can capture the dynamic fluctuations of impulsivity.

Multiplex families offer a valuable opportunity to examine whether specific features of bipolar disorder are associated with its genetic load. These families are characterized by a high prevalence of bipolar disorder and an increased burden of genetic risk variants [15]. In previous work with multiplex families, our group identified impulsivity—particularly response inhibition—as a promising endophenotype of the disorder [16].

BD involves mood fluctuations, sometimes very rapid, over a relatively short period of time. Therefore, to understand the disturbances that precede a mood change, it is particularly important to employ temporally sensitive methodologies that are not affected by retrospective biases. Recent years have highlighted the need for new types of ecologically valid measures that can provide a longitudinal assessment [17,18]. Psychiatric disorder symptoms can fluctuate rapidly, necessitating a more detailed and continuous assessment to identify critical transition points where timely intervention may be most effective. Modern Ecological Momentary Assessment (EMA) approaches, which longitudinally study individuals in their everyday natural environments using devices such as smartphones, hold promise for nuanced examination of the mechanisms of mood and impulsivity dysregulation [19,20].

Previous research using EMA approaches to explore mood and impulsivity in BD is limited, and findings are inconsistent. One pilot study compared BD patients with healthy controls (HC) on EMA measures of mood and impulsivity found no differences in mean impulsivity between the two groups, but higher variability in mean mood and impulsivity in the BD group compared to HC [21]. In another study, analysis of impulsivity measured by the EMA in BD patients found that negative, but not positive affect, predicted increases in impulsivity, which subsequently predicted decreases in positive affect [22]. Finally, Titone et al., 2022 found higher daily impulsivity in BD participants compared to HC participants. This same study demonstrated a bidirectional association between high impulsivity and high next-day negative affect and showed that impulsivity specifically predicted next-day anger and anxiety [23].

The present study investigated the concurrent and predictive relationships between impulsivity and mood in BD, using an EMA design with concurrent assessments.

## Methodology

### Setting and participants

The study was conducted at the Regional University Hospital of Malaga (Spain) between 1 July 2020 and 30 November 2021. The participant sample consisted of a convenience selection of BD participants and unaffected healthy individuals (with at least one first-degree relative with an affective disorder) members of a cohort of families with high BD prevalence from the Andalusian Bipolar Family (ABiF) study [24,25], as well as age- and sex-matched BD participants and healthy controls from the general population. Inclusion and exclusion criteria are listed in Table 1.

Thirty-four individuals were recruited to participate in the study. Four individuals dropped out before the end of the evaluation period, resulting in a final sample of 30 participants consisting of 14 males and 16 females, with a mean age of 52.60 ± 14.42 years. Participants comprised 4 groups: 1) BD participants from families (BDF, n = 8), family controls (FC, n = 6), BD ambulatory cases (BDC, n = 8), and healthy controls from the general population (HC, n = 8). Diagnoses of BD were distributed as follows: BD type I (n = 12; 5BDF,7BDC), BD type II (n = 3; 2BDF,1BDC) and BD unspecified (n = 1; 1BDF). At study entry, all participants were confirmed to be in a euthymic state. Fig 1 provides an overview of the study procedure, including recruitment, group classification, EMA data collection, and statistical analysis.

The present study was carried out with the approval of the local ethics committee (Provincial Research Ethics Committee of Málaga), and written informed consent was obtained from all participants.

### Assessments and variables

The researchers provided participants with detailed explanations of the study and its procedure. First, inclusion and exclusion criteria were checked, after which sociodemographic data were collected and a structured general health questionnaire was administered. Subsequently, in a brief training session, participants received detailed instructions on the use of the movisensXS application (movisens GmbH, Karlsruhe, Germany, https://www.movisens.com/de/produkte/movisensxs), which was used for the assessment.

Mood and impulsivity were assessed using the movisensXS application implemented on the participants' smartphones or those provided by the researchers (Samsung Galaxy J7). Participants completed the questionnaires three

**Table 1. Inclusion/exclusion criteria.**

| Groups | InclusionCriteria | ExclusionCriteria |
|---|---|---|
| All | −18+yearsold.<br>−fluently speak and read Spanish.<br>−familiar with smartphone operation. | −severe cognitive disabilities. |
| BDF | −diagnosedwith BD.<br>−member of a high density BD family.<br>−euthymic at the time of assessment. | −comorbid substance use disorder. |
| FC | −member of a high density BD family.<br>−first degree relative with affective disorder.<br>−no personal history of lifetime BD or MDD or psychotic disorder (schizophrenia, spectrum psychotic disorder). | −diagnosis of any affective disorder or substance use disorder. |
| BDC | −diagnosed with BD.<br>−not a member of one of the participating families.<br>−euthymic at the time of assessment.<br>−no first or second degree relatives with known history of BD or psychotic disorder (schizophrenia, spectrum psychotic disorder, MDD with psychotic symptoms). | −comorbid substance use disorder. |
| HC | −not a member of one of the participating families.<br>−no personal history of lifetime BD or MDD or other psychotic.<br>−disorder (schizophrenia, spectrum psychotic disorder).<br>−no first or second degree relatives with known history of BD or other psychotic disorder (schizophrenia, spectrum psychotic disorder, MDD with psychotic symptoms). | −diagnosis of any affective disorder or substance use disorder. |

BD, bipolar disorder; MDD, major depressive disorder; BDF, participants with BD who are members of multiplex BD families; FC, unaffected members of multiplex BD families; BDC, participants with BD from the general population; HC, healthy controls.

times a day (at 9:30 am, 3:30 pm and 8:30 pm) as instructed by the app. To measure mood, an item named 'General mood' was used with a visual analogue scale with a response option from 0 (downcast) to 100 (elevated). Impulsivity was assessed using The Momentary Impulsivity Scale, recommended for the EMA of the impulsivity construct [26]. Participants were asked to indicate the extent to which they felt this way in the last 15 minutes for each of the following items: 1) I said things without thinking; 2) I spent more money than I meant to; 3) I have felt impatient; 4) I made a "spur of the moment" decision. For each item, responses are selected on a Likert scale with the following options: 1: very little or not at all, 2: a little, 3: moderately, 4: quite a lot and 5: extremely. Data recorded in the movisensXS app on the smartphones were automatically uploaded to a secure server. Self-reported momentary assessments were made during a period of 6–12 weeks.

## Statistical analysis

Descriptive statistics were used to characterize the sample. For group comparisons, the Welch ANOVA test with a post-hoc Tukey test was used. Since longitudinal data with various measures over time were analyzed, mixed-effects regression models were used for analysis, with the individual as the random effect. To analyze the relationship between impulsivity and mood concurrent and lagged models were employed. We employed lagged models to assess how prior values predict subsequent changes, enabling us to capture temporal dependencies and delayed effects within the longitudinal data. Lagged effects were modeled by using the value of the independent variable at the previous time point (t-1) to predict the dependent variable at the current time point (t). Two types of models were used: 1) one in which the dependent variable was mood and the independent variable was impulsivity, and 2) another in which the dependent variable was impulsivity and the independent variable was mood. To disentangle within- and between-person effects, each time-varying predictor was decomposed into two components: (1) a person-mean centered variable representing moment-to-moment fluctuations (within-person variation), and (2) a person-level mean representing stable individual differences

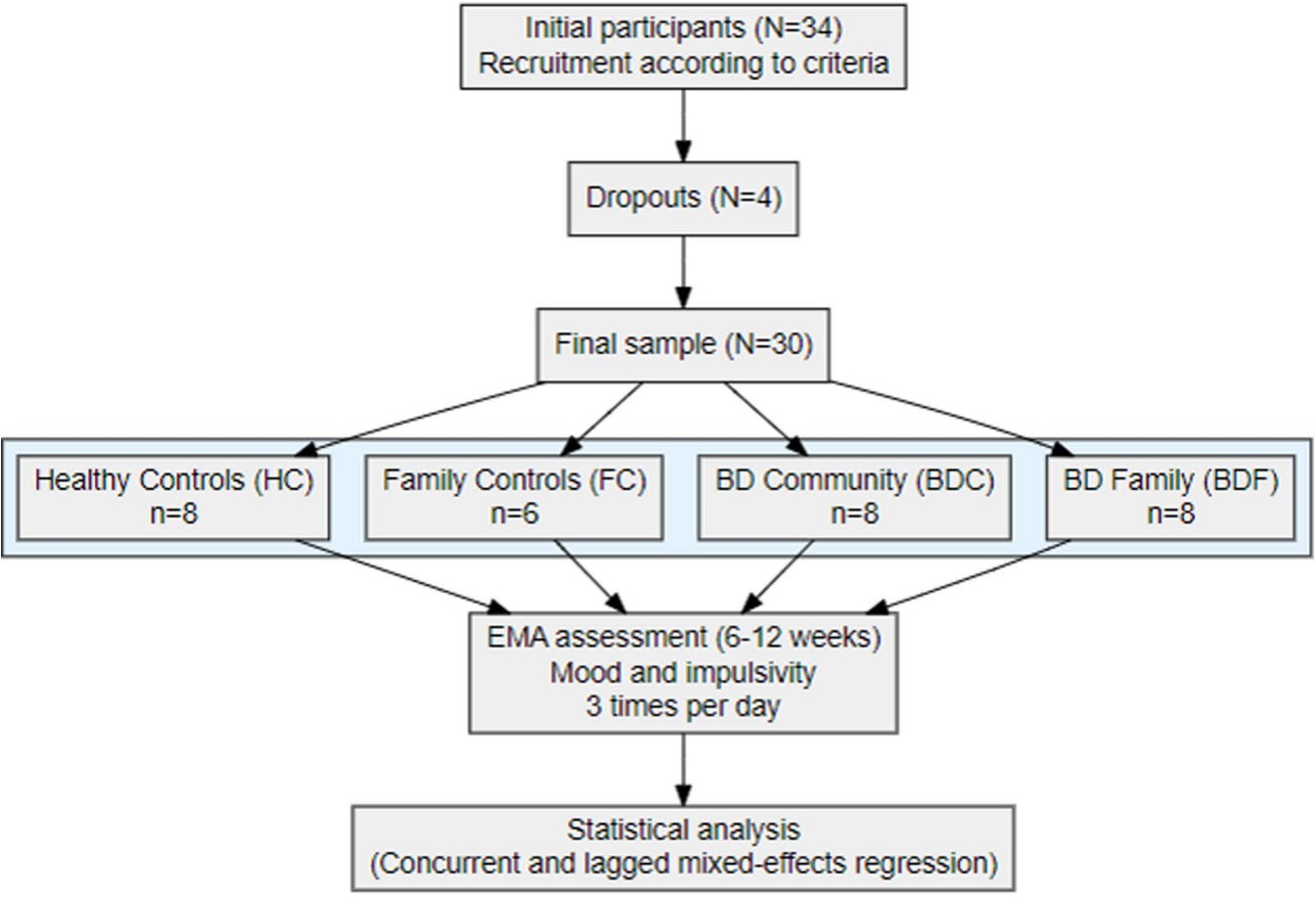

**Fig 1. Flow-chart illustrating the overall study procedure.**

(between-person variation). At first, linear models were conducted; however, due to the non-normality of the residuals (which persisted despite various transformations of the dependent variables), a dichotomous transformation of the dependent variables was carried out in order to perform mixed logistic regression models. Details regarding the fulfillment of the assumptions for the logistic models are provided in the supporting information.

A model with random intercepts and random slopes was used as it showed significantly better fit, as indicated by a lower Akaike Information Criterion. This allowed us to control for individual variability in overall mood and impulsivity, as well as in their evolution over time. Age, gender, time, presence of bipolar diagnosis and belonging to a family with multiple cases of BD were included as variables in the regression models. Belonging to a family with multiple cases of BD was included as a variable to account for potential genetic or familial influences on the relationship between impulsivity and mood, as these factors may affect the severity, frequency, or pattern of these symptoms over time. Initially, the interaction between the independent variable (time-varying) and BD diagnosis, as well as between the independent variable and belonging to families with multiple cases of BD, were included. However, due to high risk of overfitting and unstable estimates and multicollinearity (variance inflation factors > 5), we decided not to include interaction terms in the final models. The significance level was set at 0.0125 following Bonferroni correction by four comparisons (mood, impulsivity, diagnosis and belonging to families). R-Studio 2023.03.0 + 386 with R version 4.3.0.was used for the statistical analyses.

## Results

The sociodemographic and clinical characteristics of the sample, as well as the differences between groups, are shown in Table 2. There were significant differences in average mood between the groups (F = 4.81, *p* = .017), with differences observed between BDF and HC (Diff = −31.70, *p* = .001) and BDC and HC (Diff = −25.94, *p* = .007), but no significant differences when compared with FC. No significant differences between groups in average impulsivity were found. Fig 2 shows mood and impulsivity scores over time across the four groups.

**Table 2. Sociodemographic and clinical characteristics of the sample.**

| Variables | Total | BDF (N = 8) | BDC (N = 8) | FC (N = 6) | HC (N = 8) | *p* |
|---|---|---|---|---|---|---|
| Age, M ± SD | 52.60 ± 14.42 | 55.88 ± 12.79 | 51.50 ± 8.70 | 53.00 ± 17.92 | 50.13 ± 19.30 | .886 |
| Gender, Female, N (%) | 17 (56.67) | 3 (37.55) | 4 (50) | 4 (66.67) | 6 (75) | .443 |
| Mood*, M ± SD | 57.52 ± 18.66 | 45.37 ± 16.72 | 51.13 ± 10.12 | 56.16 ± 10.49 | 77.08 ± 18.09 | .017 |
| Impulsivity**, M ± SD | 5.64 ± 2.02 | 5.95 ± 2.83 | 6.96 ± 2.04 | 4.73 ± 0.48 | 4.72 ± 0.855 | .060 |

Note: BDF: Participants from families diagnoses of bipolar disorders. BDC: Participants from the community diagnoses of bipolar disorders. FC: Family controls. HC: Healthy controls from the general population.

*Average mood in all registers per participant. Differences between BDF vs. HC: Diff = −31.70, *p* = .001, BDC vs HC: Diff. = −25.74, *p* = .007.

** Average impulsivity in all registers per participant.

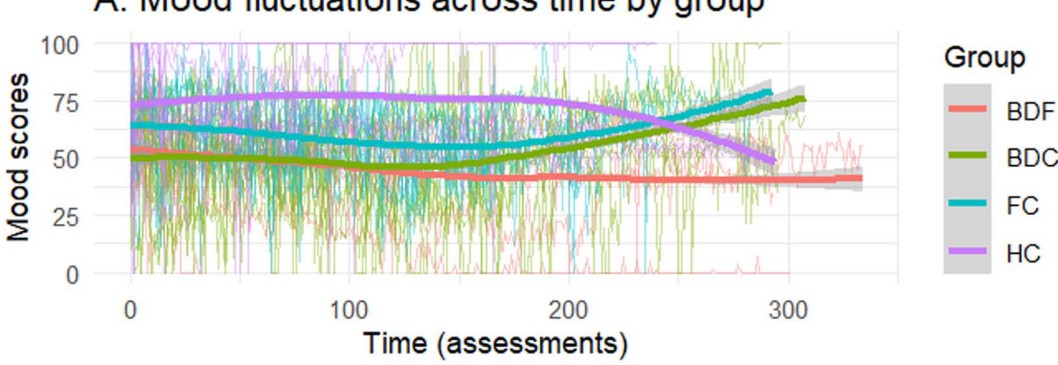

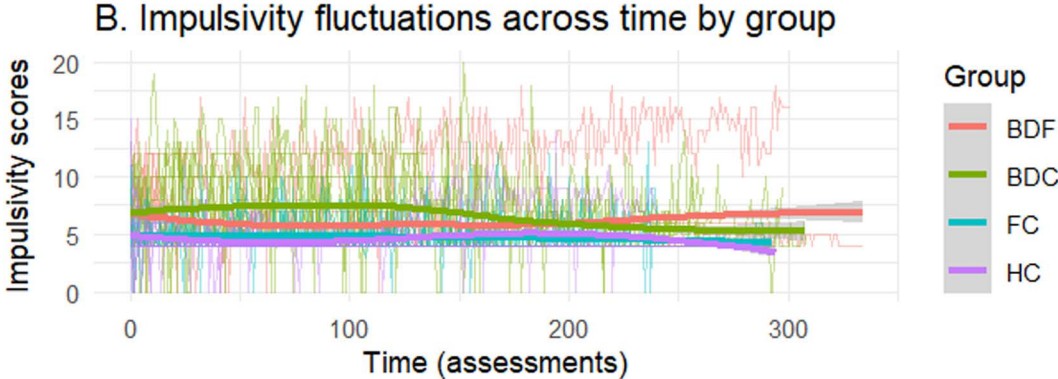

**Fig 2. Mood and impulsivity fluctuations across time by group.**

## Mood as dependent variable

In the concurrent model, mean impulsivity during the assessment period was related to lower mood (OR=0.937, *p*=0.001) and variation from the individual's mean impulsivity was related to lower mood (OR=0.937, *p*=0.001). In the lagged model, variation with respect to the individual's mean impulsivity was related to lower mood (OR= 0.939, *p*=0.002). The diagnosis of BD and belonging to multiplex families were near to significance in concurrent and lagged models but not reach the threshold. More information on the conducted analyses is provide in Table 3. Fig 3 shows the predicted probability of experiencing high mood based on momentary impulsivity (centered within individuals and lagged by one assessment).

Table 3. Concurrent and lagged logistic mixed regression model using mood assessed by EMA as the dependent variable.

| Models (Dependent variable: Mood) | Concurrent Model | | | | Lagged Model | | | |
|---|---|---|---|---|---|---|---|---|
| Fixed Effects | SE | z | OR | p | SE | z | OR | p |
| Impulsivity (time varying) | 0.020 | −3.201 | 0.937 | 0.001 | 0.020 | −3.084 | 0.939 | 0.002 |
| Average Impulsivity (time invaring) | 0.191 | −3.454 | 0.518 | <0.001 | 0.193 | −3.398 | 0.518 | <0.001 |
| Bipolar diagnosis | 0.753 | −1.641 | 0.290 | 0.101 | 0.764 | −1.682 | 0.276 | 0.092 |
| Multiplex family belonging | 0.710 | −1.992 | 0.243 | 0.046 | 0.715 | −2.037 | 0.233 | 0.042 |
| Time | 0.004 | −1.112 | 0.995 | 0.266 | 0.004 | −1.288 | 0.994 | 0.198 |
| Age | 0.024 | −1.274 | 0.969 | 0.203 | 0.025 | −1.275 | 0.969 | 0.202 |
| Gender | 0.709 | 1.379 | 2.659 | 0.168 | 0.719 | 1.351 | 2.643 | 0.177 |

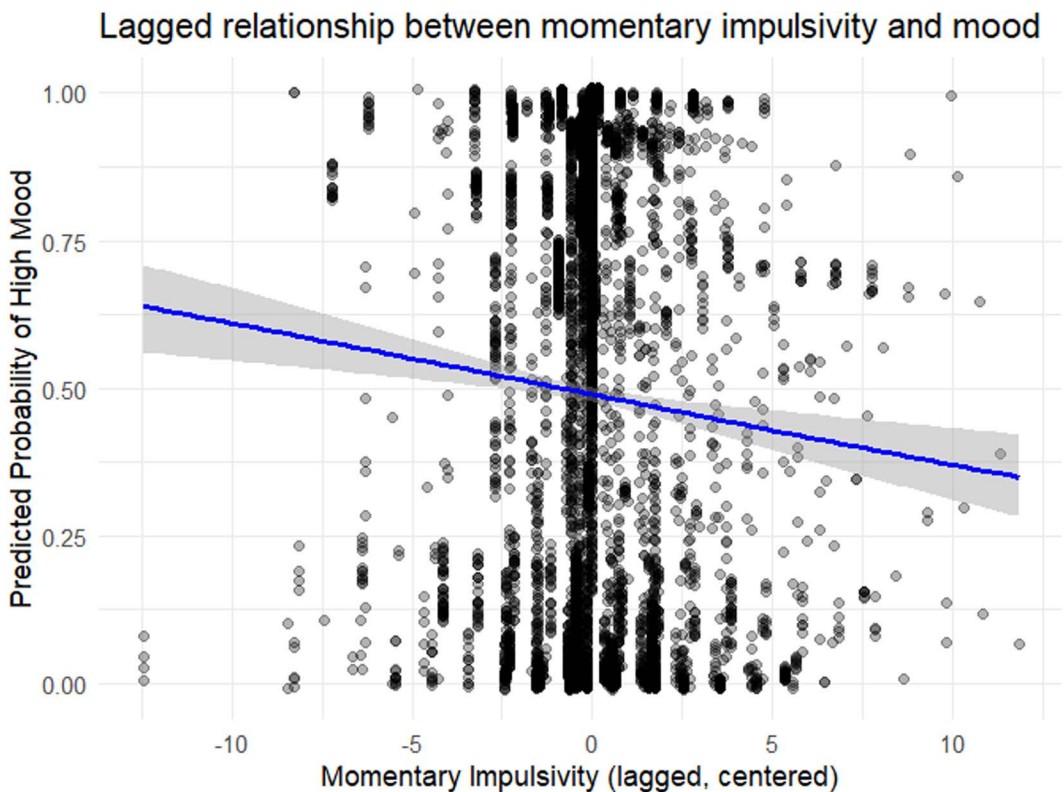

**Fig 3. Lagged relationship between momentary impulsivity and mood.**

The blue line represents the logistic regression fit derived from the mixed-effects model, with a 95% confidence interval shaded in gray.

## Impulsivity as dependent variable

Similar to the previous model, in the concurrent model, lower mean mood during the assessment period was related to higher impulsivity (OR=0.940, $p < 0.001$) and variation from the individual's mean mood was near to reach significance (OR=0.094, $p = 0.015$). However, in the lagged model, variation with respect to the individual's mean mood was not related to subsequent change in impulsivity (OR=0.096, $p = 0.135$). The belonging to multiplex families was near to significance in concurrent and lagged models but not reach the threshold. More information can be found in Table 4.

## Discussion

The study found a significant negative association between impulsivity and concurrent mood at the EMA. Furthermore, a significant relationship was found between prior impulsivity and mood at the next assessment independent of diagnosis, whereas the reverse did not occur, no significant association was found between mood at the EMA and impulsivity at the subsequent assessment. Those with BD were also found to have a downcast mood in comparison with controls but not with controls from the families. These results suggest that impulsivity could be a risk factor for subsequent downcast mood, although further studies are needed to confirm this hypothesis. To the best of our knowledge, no other study has conducted an assessment using EMA to investigate the concurrent and predictive relationships between impulsivity and mood in families with high prevalence for BD. Hence, our results represent only a preliminary examination about relationship between impulsivity and mood on ecological momentary assessments (EMA).

These results are consistent with previous research that has shown a relationship between impulsivity and negative mood [27–29] in both clinical [30] and non-clinical populations [31,32]. This relationship has also been found when studies have been conducted using ecological intensive measurement [33]. The possibility of a shared serotonergic pathway between the two conditions has been raised to explain this relationship [34].

Regarding the temporal relationships between impulsivity and mood, the reported findings align with studies in samples without BD that found a strong association between daily impulsivity and negative affect [27–29]. However, our findings differ slightly from two previous studies [22,33]. While Depp and colleagues [22] found that negative affect predicted impulsivity, which in turn predicted a decrease in positive affect, we found a significant unidirectional relationship from impulsivity to mood. For its part, Titone et al. [33] found a bidirectional relationship between impulsivity and negative affect, and no significant relationship between positive affect and impulsivity.

These mixed findings could be attributed to the heterogeneity of the studies. In our study cohort during follow-up, all participants were confirmed to be in a euthymic state, representing non-severe cases due to the exclusion of the most unstable cases. In contrast, in Titone's [33] study, although some participants were in full remission, most were

**Table 4. Concurrent and lagged logistic mixed regression model using impulsivity as the dependent variable assessed by EMA.**

| Models (Dependent variable: Impulsivity) | Concurrent model | | | | Lagged Model | | | |
|---|---|---|---|---|---|---|---|---|
| Fixed Effects | SE | z | OR | p | SE | z | OR | p |
| Mood (time varying) | 0.002 | −2.424 | 0.994 | 0.015 | 0.002 | −1.495 | 0.996 | 0.135 |
| Average Mood (time invarying) | 0.018 | −3.488 | 0.940 | <0.001 | 0.018 | −3.462 | 0.939 | <0.001 |
| Bipolar diagnosis | 0.624 | −0.557 | 0.706 | 0.577 | 0.643 | −0.616 | 0.673 | 0.538 |
| Multiplex family belonging | 0.536 | −1.952 | 0.351 | 0.051 | 0.552 | −1.996 | 0.332 | 0.046 |
| Time | 0.003 | −1.079 | 0.997 | 0.280 | 0.003 | −0.931 | 0.997 | 0.352 |
| Age | 0.018 | −2.323 | 0.959 | 0.020 | 0.018 | −2.110 | 0.962 | 0.035 |
| Gender | 0.529 | −0.013 | 0.993 | 0.989 | 0.543 | −0.095 | 0.950 | 0.925 |

symptomatic at the time of assessment and met criteria for a (hypo)manic or depressive episode. Similarly, in Depp's [22] study, at the time of baseline assessment, the sample, on average, exhibited a mild level of depression and manic symptoms. In this context, it would be interesting to clarify whether this bidirectional interaction might have been influenced by depressive or manic mood states. Moreover, considerable heterogeneity was observed in the assessment scales used across studies, where the items and response options employed to evaluate impulsivity and mood varied notably.

On the other hand, although there is a large body of evidence showing that impulsivity plays a role in BD [1,16], the present study did not find a significant relationship between BD diagnosis and belonging to multiplex families and impulsivity. This result could be influenced by the small size of the study groups, as both diagnosis and family membership approached significance in the concurrent and lagged models.

These findings provide preliminary evidence that impulsivity may influence negative mood, potentially acting as a risk factor, although this relationship remains uncertain. Further research is needed to explore the distinct facets of impulsivity, its temporal dynamics, and its predictive value regarding mood over extended periods and across diverse clinical populations. Nevertheless, the present study offers an initial step toward understanding the role of impulsivity in mood regulation.

## Limitations

To interpret these results, the limitations of the study must be considered. The main limitation of the study is the small sample size, with only 30 participants distributed across four groups. This limited number of participants may have reduced the statistical power to detect subtle effects, increased the risk of Type II errors, and limited the generalizability of the findings. Although the intensive, repeated-measures design provided a large number of momentary assessments, the individual-level sample size remains small. No a priori power analysis was conducted, as the study was designed as an exploratory investigation. Also, the EMA measure of impulsivity relied on four self-reported items from the Momentary Impulsivity Scale. While this instrument has been recommended for use in EMA contexts, it has not yet been fully validated in clinical populations, and may not capture the full complexity of the impulsivity construct. Therefore, results related to momentary impulsivity should be interpreted with caution and considered preliminary. Similarly, mood was assessed using a single-item visual analogue scale ranging from 0 to 100. Although this approach facilitated repeated assessments with minimal participant burden, it may have limited the multidimensionality of the mood data. The use of more comprehensive scales (e.g., PANAS or PHQ-9) might have provided a more nuanced picture of affective states. Moreover, the exclusion of participants with substance use disorders reduced potential confounding but may limit generalizability, as this subgroup is common in BD and may show different patterns of impulsivity and mood.

## Conclusions

Our results indicate that based in intensive longitudinal data a relationship between impulsivity and concurrent negative mood in those with BD and a relationship between previous impulsivity and negative mood at the next assessment regardless of diagnosis were found. These results offer a preliminary contribution to understanding the complex interactions between BD, genetic load, impulsivity, and mood. While our findings point to the possibility that impulsivity may be a relevant factor in mood regulation among individuals with bipolar disorder, further research with larger and more diverse samples is necessary to clarify this relationship and evaluate its potential implications for clinical interventions.

## Supporting information

**S1 Appendix. Diagnostics and Assumption Checks for the logistic mixed models.**
(DOCX)

## Acknowledgments

We would like to thank all the participants for their commitment to and participation in the study.

## Author contributions

**Conceptualization:** Lea Sirignano, Jerome C. Foo, Fabian Streit, Josef Frank, Stephanie H. Witt, Marcella Rietschel, Fermin Mayoral-Cleries, Berta Moreno-Küstner, Jose Guzman-Parra.

**Data curation:** Almudena Ramírez-Martín, Lea Sirignano, Jose Guzman-Parra.

**Formal analysis:** Jose Guzman-Parra.

**Investigation:** Almudena Ramírez-Martín, Lea Sirignano, Jerome C. Foo, Fabian Streit, Josef Frank, Stephanie H. Witt, Marcella Rietschel, Fermin Mayoral-Cleries, Berta Moreno-Küstner, Jose Guzman-Parra.

**Methodology:** Almudena Ramírez-Martín, Lea Sirignano, Jerome C. Foo, Josef Frank, Jose Guzman-Parra.

**Project administration:** Almudena Ramírez-Martín, Lea Sirignano, Marcella Rietschel.

**Resources:** Marcella Rietschel, Fermin Mayoral-Cleries.

**Supervision:** Lea Sirignano, Jerome C. Foo, Fabian Streit, Marcella Rietschel, Berta Moreno-Küstner.

**Writing – original draft:** Almudena Ramírez-Martín, Jose Guzman-Parra.

**Writing – review & editing:** Lea Sirignano, Jerome C. Foo, Fabian Streit, Josef Frank, Stephanie H. Witt, Marcella Rietschel, Fermin Mayoral-Cleries, Berta Moreno-Küstner.

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
