## [Decision Letter · Decision Letter 0]

PONE-D-24-52633The relationships between impulsivity and mood in bipolar disorder: An ecological momentary assessment studyPLOS ONE

Dear Dr. Guzman-Parra,

Thank you for submitting your manuscript to PLOS ONE. After careful consideration, we feel that it has merit but does not fully meet PLOS ONE’s publication criteria as it currently stands. Therefore, we invite you to submit a revised version of the manuscript that addresses the points raised during the review process.

We look forward to receiving your revised manuscript.

Kind regards,

Mehdi Rezaei

Academic Editor

PLOS ONE

2. Thank you for stating the following financial disclosure:  [The study was supported by the German Research Foundation (DFG) and the German Federal Ministry of Education and Research (BMBF) through the Integrated Network IntegraMent under the auspices of the e:Med programme [grants 01ZX1314A to MMN and SC; 01ZX1314G and 01ZX1614G to MR], through “ASD-Net” [grant 01EE1409C to MR and SHW]; by ERA-NET NEURON through “SynSchiz - Linking synaptic dysfunction to disease mechanisms in schizophrenia – a multilevel investigation“ [grant 01EW1810 to MR] and “EMBED - impact of Early life MetaBolic and psychosocial strEss on susceptibility to mental Disorders; from converging epigenetic signatures to novel targets for therapeutic intervention” [grant 01EW1904]; “FOR2107” [grants RI908/11-2 to MR; NO246/10-2 to MMN and WI3439/3-2 to SHW]; by the Andalusian regional Health and Innovation Government [grants PI-0060-2017]. MMN is a member of the DFG-funded cluster of excellence ImmunoSensation.].  Please state what role the funders took in the study.  If the funders had no role, please state: "The funders had no role in study design, data collection and analysis, decision to publish, or preparation of the manuscript." If this statement is not correct you must amend it as needed. Please include this amended Role of Funder statement in your cover letter; we will change the online submission form on your behalf.

3. In the online submission form, you indicated that [The data underlying the results presented in the study are available from corresponding author.]. All PLOS journals now require all data underlying the findings described in their manuscript to be freely available to other researchers, either 1. In a public repository, 2. Within the manuscript itself, or 3. Uploaded as supplementary information. This policy applies to all data except where public deposition would breach compliance with the protocol approved by your research ethics board. If your data cannot be made publicly available for ethical or legal reasons (e.g., public availability would compromise patient privacy), please explain your reasons on resubmission and your exemption request will be escalated for approval.

Additional Editor Comments:

The manuscript, according to the reviewer's opinion, requires major revisions. Please carefully address all the suggested changes.

Reviewers' comments:

Reviewer's Responses to Questions

**Comments to the Author**

1. Is the manuscript technically sound, and do the data support the conclusions?

Reviewer #1: Partly

Reviewer #2: Yes

2. Has the statistical analysis been performed appropriately and rigorously? 

Reviewer #1: Yes

Reviewer #2: Yes

3. Have the authors made all data underlying the findings in their manuscript fully available?

Reviewer #1: No

Reviewer #2: Yes

4. Is the manuscript presented in an intelligible fashion and written in standard English?

Reviewer #1: Yes

Reviewer #2: Yes

5. Review Comments to the Author

Reviewer #1: Review Summary

This study explores the relationship between impulsivity and mood in individuals with bipolar disorder (BD) using ecological momentary assessment (EMA). By capturing real-time fluctuations in impulsivity and mood, the study provides valuable insights into their dynamic interplay. Additionally, the inclusion of multiplex family participants offers an interesting perspective on potential genetic and environmental influences. The use of mixed-effects regression models to examine both concurrent and lagged effects further strengthens the methodological approach.

Despite these strengths, several critical issues need to be addressed before the manuscript can be considered for publication.The most pressing concern is the potential overfitting of the statistical models given the small sample size (n=30).The use of complex mixed-effects regression models, particularly the lagged analyses, may not be fully appropriate for a dataset of this size and could lead to unstable or misleading results. The manuscript would also benefit from better data visualization, a more thorough discussion of the findings in relation to existing literature, and greater clarity regarding data-sharing practices.

Strengths

This study is valuable in several ways. First, it addresses an important clinical issue by examining the relationship between impulsivity and mood instability in BD, which has significant implications for treatment and symptom management. Second, the use of EMA allows for real-time data collection, reducing recall bias and improving the ecological validity of the findings. Third, the inclusion of multiplex family participants adds a novel dimension to the study, enabling an exploration of potential genetic and familial influences on impulsivity and mood regulation. Finally, the application of mixed-effects regression models is methodologically sophisticated and allows for a nuanced understanding of both concurrent and time-lagged relationships.

Areas for Improvement

1. Addressing Potential Overfitting in Statistical Models

One of the primary concerns with this study is the potential overfitting of the statistical models. The use of mixed-effects regression models, particularly with a small sample size of only 30 participants, may lead to unreliable parameter estimates and reduced generalizability. The inclusion of multiple predictors in these models, combined with the complexity of lagged analyses, increases the risk of overfitting and statistical instability.

To address this, I strongly recommend that the authors assess the risk of overfitting by examining model assumptions, variance inflation factors, and potential multicollinearity. Additionally, alternative statistical approaches should be considered to validate the robustness of the findings. For instance, conducting repeated measures ANOVA or generalized linear mixed models (GLMM) may provide a more appropriate way to analyze the data given the sample size. If the authors choose to retain the current modeling approach, they should justify why mixed-effects models are necessary and discuss the limitations of applying such complex methods with a small dataset.

Suggested Revisions:

- Assess overfitting risks and confirm model assumptions.

- Conduct additional analyses using simpler statistical models and compare results.

- Justify the use of mixed-effects models and acknowledge their limitations.

2. Enhancing Data Visualization

The manuscript would benefit from clearer visual representation of the study’s methodology and key findings.Currently, there is no flowchart outlining participant recruitment, group classification, and study procedures,making it difficult to follow the research design. Furthermore, the results section lacks figures depicting the impulsivity-mood relationship,which would greatly aid in interpretation.

To improve clarity, I recommend including:

- A flowchart summarizing participant selection, group categorization, and data collection.

- Scatter plots with regression lines to illustrate the relationship between impulsivity and mood.

- Line graphs to show how impulsivity and mood fluctuate over time across different groups.

Suggested Revisions:

- Add a flowchart illustrating the study design.

- Include scatter plots and line graphs to enhance the presentation of results.

3. Strengthening the Discussion: Comparison with Existing Literature

The discussion section provides a solid interpretation of the results but could be further strengthened by placing the findings in a broader research context. Specifically, the study reports no significant difference in impulsivity between multiplex BD patients and non-multiplex BD patients,despite previous research suggesting otherwise. However, the manuscript does not sufficiently explain this discrepancy.

To enhance the discussion, I suggest exploring possible reasons why multiplex BD patients did not exhibit greater impulsivity.Factors such as sample size limitations, differences in assessment methods, or potential protective influences in multiplex families should be considered. Additionally, the study’s findings should be compared more extensively with existing research on impulsivity in BD.

Suggested Revisions:

- Discuss why the study’s findings differ from previous research.

- Consider alternative explanations (e.g., sample size, protective factors).

- Strengthen comparisons with existing literature on BD and impulsivity.

4. Clarifying Data Availability Statement

PLOS ONE encourages open data-sharing practices whenever possible. While the manuscript states that data are available upon request, providing access to the dataset through a public repository (e.g., OSF, Dryad) would be preferable. If ethical or legal constraints prevent full data sharing, a brief clarification in the manuscript would be helpful.

Suggested Revisions:

- If feasible, upload the dataset to a public repository and provide a DOI.

- If data-sharing restrictions apply, briefly explain the reason in the manuscript.

Reviewer #2: Major Concerns

1. Sample Size & Generalizability

o The study includes only 30 participants across four groups, which is quite small for making strong conclusions. The authors should acknowledge this limitation more explicitly and discuss how it affects the generalizability of the findings.

o Were any power analyses conducted to determine whether the sample size was sufficient to detect significant effects?

2. Measurement of Impulsivity and Mood

o The ecological momentary assessment (EMA) measure of impulsivity relies on four self-reported items that have not been fully validated. This should be more explicitly acknowledged as a limitation.

o Mood was measured using a single-item scale (0-100). While EMA is useful for momentary assessment, using a more comprehensive mood scale (e.g., PANAS, PHQ-9) might provide a richer picture.

3. Statistical Model Clarity

o The mixed-effects regression models used are appropriate, but explaining how lagged effects were handled could be clearer.

o It would be helpful to clarify whether the results controlled for individual variability in impulsivity and mood trends over time.

4. Direction of Causality

o The study finds that impulsivity predicts lower mood at the next assessment, but mood does not predict impulsivity. This contradicts some prior EMA studies (e.g., Titone et al., 2022). The authors should discuss alternative explanations, such as possible mediators (e.g., stress or sleep quality).

o Reverse causality (mood influencing impulsivity) should be discussed more thoroughly.

Minor Concerns

1. Clarify Inclusion/Exclusion Criteria

o The exclusion of individuals with substance use disorders is reasonable but may have removed an important subgroup in BD. A brief discussion on how this impacts the results would be helpful.

2. More Balanced Discussion of Findings

o The discussion focuses primarily on impulsivity as a risk factor for mood changes. However, prior studies suggest a bidirectional relationship. The authors should provide a more nuanced interpretation of how their findings fit into existing literature.

3. Consistency in Terminology

o The term “multiplex families” is used throughout but is not clearly defined in the introduction. Adding a brief definition (e.g., “families with multiple first-degree relatives diagnosed with BD”) would be helpful.

4. References and Citations

o Some references (e.g., Titone et al., 2022; Deep et al., 2016) are mentioned, but their findings could be contrasted more explicitly with the present results.

Overall Recommendation

The study is well-structured and addresses an important topic in BD research. However, given the small sample size and limitations of impulsivity measurement, the conclusions should be more cautious. A more explicit discussion of alternative explanations and prior inconsistent findings would strengthen the manuscript.

6. PLOS authors have the option to publish the peer review history of their article (what does this mean? ). If published, this will include your full peer review and any attached files.

**Do you want your identity to be public for this peer review?** For information about this choice, including consent withdrawal, please see our Privacy Policy .

Reviewer #1: No

Reviewer #2: No

---

## [Author Response · Author response to Decision Letter 1]

23 Apr 2025

Reviewer #1: Review Summary

This study explores the relationship between impulsivity and mood in individuals with bipolar disorder (BD) using ecological momentary assessment (EMA). By capturing real-time fluctuations in impulsivity and mood, the study provides valuable insights into their dynamic interplay. Additionally, the inclusion of multiplex family participants offers an interesting perspective on potential genetic and environmental influences. The use of mixed-effects regression models to examine both concurrent and lagged effects further strengthens the methodological approach.

Despite these strengths, several critical issues need to be addressed before the manuscript can be considered for publication.The most pressing concern is the potential overfitting of the statistical models given the small sample size (n=30).The use of complex mixed-effects regression models, particularly the lagged analyses, may not be fully appropriate for a dataset of this size and could lead to unstable or misleading results. The manuscript would also benefit from better data visualization, a more thorough discussion of the findings in relation to existing literature, and greater clarity regarding data-sharing practices.

Strengths

This study is valuable in several ways. First, it addresses an important clinical issue by examining the relationship between impulsivity and mood instability in BD, which has significant implications for treatment and symptom management. Second, the use of EMA allows for real-time data collection, reducing recall bias and improving the ecological validity of the findings. Third, the inclusion of multiplex family participants adds a novel dimension to the study, enabling an exploration of potential genetic and familial influences on impulsivity and mood regulation. Finally, the application of mixed-effects regression models is methodologically sophisticated and allows for a nuanced understanding of both concurrent and time-lagged relationships.

Areas for Improvement

1. Addressing Potential Overfitting in Statistical Models

One of the primary concerns with this study is the potential overfitting of the statistical models. The use of mixed-effects regression models, particularly with a small sample size of only 30 participants, may lead to unreliable parameter estimates and reduced generalizability. The inclusion of multiple predictors in these models, combined with the complexity of lagged analyses, increases the risk of overfitting and statistical instability.

To address this, I strongly recommend that the authors assess the risk of overfitting by examining model assumptions, variance inflation factors, and potential multicollinearity. Additionally, alternative statistical approaches should be considered to validate the robustness of the findings. For instance, conducting repeated measures ANOVA or generalized linear mixed models (GLMM) may provide a more appropriate way to analyze the data given the sample size. If the authors choose to retain the current modeling approach, they should justify why mixed-effects models are necessary and discuss the limitations of applying such complex methods with a small dataset.

Suggested Revisions:

- Assess overfitting risks and confirm model assumptions.

- Conduct additional analyses using simpler statistical models and compare results.

- Justify the use of mixed-effects models and acknowledge their limitations.

Thank you very much for your valuable suggestions. We greatly appreciate your dedication and helpful recommendations aimed at improving our analyses and enhancing the robustness of the study. The reviewer correctly identified important concerns related to the modeling approach. Upon examining model assumptions, we detected heteroscedasticity in the residuals. Consequently, we attempted transformations of the dependent variable; however, these problems persisted. To address this issue, we ultimately chose to transform the dependent variable into a dichotomous variable based on the median and performed mixed logistic regression analyses.

Additionally, we decided to remove interaction terms because adequately testing interactions would require a substantially larger sample size—far beyond our current sample—which would otherwise lead to a high risk of overfitting and unstable estimates. We have ensured that all assumptions for the revised regression models are adequately met and have provided evidence in the supplementary materials. The main findings remain consistent with those obtained from the previous models. Given that our revised logistic mixed regression models based on a dichotomized dependent variable provide stable and consistent results, we believe that additional sensitivity analyses, as proposed, would not satisfy necessary model assumptions nor provide further meaningful insights. We have updated all tables and revised the statistical analysis section accordingly.

2. Enhancing Data Visualization

The manuscript would benefit from clearer visual representation of the study’s methodology and key findings.Currently, there is no flowchart outlining participant recruitment, group classification, and study procedures,making it difficult to follow the research design. Furthermore, the results section lacks figures depicting the impulsivity-mood relationship,which would greatly aid in interpretation.

To improve clarity, I recommend including:

- A flowchart summarizing participant selection, group categorization, and data collection.

- Scatter plots with regression lines to illustrate the relationship between impulsivity and mood.

- Line graphs to show how impulsivity and mood fluctuate over time across different groups.

Suggested Revisions:

- Add a flowchart illustrating the study design.

- Include scatter plots and line graphs to enhance the presentation of results.

Thank you very much for the suggestion. The reviewer is right in pointing out that the study lacked the visual component that helps readers better understand the article and clarifies the meaning of the data. Following the reviewer’s recommendation, we have incorporated the suggested graphs as Figure 1, Figure 2, and Figure 3.

3. Strengthening the Discussion: Comparison with Existing Literature

The discussion section provides a solid interpretation of the results but could be further strengthened by placing the findings in a broader research context. Specifically, the study reports no significant difference in impulsivity between multiplex BD patients and non-multiplex BD patients,despite previous research suggesting otherwise. However, the manuscript does not sufficiently explain this discrepancy.

To enhance the discussion, I suggest exploring possible reasons why multiplex BD patients did not exhibit greater impulsivity.Factors such as sample size limitations, differences in assessment methods, or potential protective influences in multiplex families should be considered. Additionally, the study’s findings should be compared more extensively with existing research on impulsivity in BD.

Suggested Revisions:

- Discuss why the study’s findings differ from previous research.

- Consider alternative explanations (e.g., sample size, protective factors).

- Strengthen comparisons with existing literature on BD and impulsivity.

Thank you very much for this insightful comment. We agree that the discrepancy between our findings and previous research on impulsivity in multiplex BD patients required further elaboration. Following your suggestion, we have revised the discussion section to explore possible explanations, including the limited sample size, potential protective factors in multiplex families, and methodological differences in the assessment of impulsivity. We have also expanded the comparison with existing literature on impulsivity in BD, aiming to provide a more comprehensive interpretation of our results within the broader research context.

4. Clarifying Data Availability Statement

PLOS ONE encourages open data-sharing practices whenever possible. While the manuscript states that data are available upon request, providing access to the dataset through a public repository (e.g., OSF, Dryad) would be preferable. If ethical or legal constraints prevent full data sharing, a brief clarification in the manuscript would be helpful.

Suggested Revisions:

- If feasible, upload the dataset to a public repository and provide a DOI.

- If data-sharing restrictions apply, briefly explain the reason in the manuscript.

We have reconsidered the possibility of sharing the data. Our initial concern was related to the potential risk to participant confidentiality, particularly regarding the involvement of families. However, since the dataset does not contain any pedigree information and only indicates group membership at a general level, we believe the data can be shared without compromising confidentiality. Please find the link here: https://osf.io/6qsbx/

Thank you for submitting to PLOS One. Here are some suggestions for the manuscript titled "The relationships between impulsivity and mood in bipolar disorder: An ecological momentary assessment study."

Major Concerns

1. Sample Size & Generalizability

◦ The study includes only 30 participants across four groups, which is quite small for making strong conclusions. The authors should acknowledge this limitation more explicitly and discuss how it affects the generalizability of the findings.

◦ Were any power analyses conducted to determine whether the sample size sufficiently detected significant effects?

We appreciate the reviewer’s thoughtful comment regarding the sample size. We agree that the small number of participants (n=30), despite the intensive longitudinal design, constitutes a key limitation of the present study. We have now revised the Limitations section of the manuscript to more explicitly acknowledge that the limited sample may reduce statistical power, increase the risk of Type II errors, and constrain the generalizability.

Additionally, we clarify that no a priori power analysis was conducted. This study was designed as an exploratory investigation of within-person mood-impulsivity dynamics using ecological momentary assessment (EMA) in a rare and complex sample that includes participants from multiplex bipolar disorder families. While the dense momentary data provided valuable insights, we agree that future studies with larger and more diverse samples will be essential to confirm and extend our findings.

2. Measurement of Impulsivity and Mood

◦ The ecological momentary assessment (EMA) measure of impulsivity relies on four self-reported items that have not been fully validated. This should be more explicitly acknowledged as a limitation.

◦ Mood was measured using a single-item scale (0-100). While EMA is useful for momentary assessment, using a more comprehensive mood scale (e.g., PANAS, PHQ-9) might provide a richer picture.

We thank the reviewer for highlighting this important point. We agree that the EMA measure of impulsivity, although based on the Momentary Impulsivity Scale recommended for EMA contexts, consists of four self-reported items that have not been fully validated in clinical populations. We have now acknowledged this more explicitly as a limitation in the manuscript.

Similarly, we recognize that mood was assessed using a single-item visual analogue scale ranging from 0 to 100, which may not fully capture the complexity of affective states. While the use of brief measures is often necessary in EMA to minimize participant burden and maintain compliance, we acknowledge that more comprehensive scales (e.g., PANAS or PHQ-9) might provide a richer and more nuanced assessment of mood. This point has also been added to the Limitations section to provide a more balanced view of the methodological constraints.

3. Statistical Model Clarity

◦ The mixed-effects regression models used are appropriate, but explaining how lagged effects were handled could be clearer.

◦ It would be helpful to clarify whether the results controlled for individual variability in impulsivity and mood trends over time.

We thank the reviewer for this helpful suggestion. We have now clarified in the Statistical analysis section how lagged effects were modeled using prior assessments (t-1), and how time-varying predictors were decomposed to distinguish within- and between-person effects. Furthermore, we confirm that our models included random intercepts and random slopes for time at the participant level, which allowed us to control for individual differences in mood and impulsivity trajectories over time.

4. Direction of Causality

◦ The study finds that impulsivity predicts lower mood at the following assessment, but mood does not predict impulsivity. This contradicts some prior EMA studies (e.g., Titone et al., 2022). The authors should discuss alternative explanations, such as possible mediators (e.g., stress or sleep quality).

◦ Reverse causality (mood influencing impulsivity) should be discussed more thoroughly.

We appreciate the reviewer's valuable suggestions and agree that the study would benefit from further discussion on this aspect.

In the new version of the manuscript, we have incorporated a more detailed explanation of the discrepancies between our findings and those of previous studies, considering possible factors that might have contributed to these mixed results, such methodological differences, participants' clinical status, and small sample size.

Minor Concerns

1. Clarify Inclusion/Exclusion Criteria

◦ The exclusion of individuals with substance use disorders is reasonable but may have removed an essential subgroup in BD. A brief discussion on how this impacts the results would be helpful.

Thank you for this observation. We have added a brief note in the Limitations section acknowledging that excluding individuals with substance use disorders may limit the generalizability of the findings, as this subgroup is often present in BD and may present distinct clinical features.

2. More Balanced Discussion of Findings

◦ The discussion focuses primarily on impulsivity as a risk factor for mood changes. However, prior studies suggest a bidirectional relationship. The authors should provide a more nuanced interpretation of how their findings fit into existing literature.

We thank the reviewer for this observation.

In the revised version of the manuscript, we have expanded the discussion to contextualise our results within the existing body of literature that has found bidirectional relationships between impulsivity and mood. Furthermore, we have emphasised that, although our results show a unidirectional association from impulsivity to mood, this interpretation should be considered preliminary and subject to verification in future research with larger samples and diverse methodological designs.

3. Consistency in Terminology

◦ The term “multiplex families” is used throughout but is not clearly defined in the introduction. Adding a brief definition (e.g., “families with multiple first-degree relatives diagnosed with BD”) would be helpful.

We thank the reviewer for this helpful comment. We have now added a brief definition of "multiplex families" in the Introduction, clarifying that these are families with a high prevalence of bipolar disorder and an increased burden of genetic risk variants. We have also referenced prior work from our group using this type of sample to provide further context.

4. References and Citations

◦ Some references (e.g., Titone et al., 2022; Deep et al., 2016) are mentioned, but their findings could be contrasted more explicitly with the present results.

We thank the reviewer for this valuable observation. We have revised and expanded the discussion section to more explicitly contrast the findings of previous studies, such as those of Titone et al. (2022) and Depp et al. (2016), with the results obtained in the present study.

This more detailed comparison allows us to place our findings within the framework of the existing literature and highlights the need for further research on the relationship between impulsivity and mood

---

## [Decision Letter · Decision Letter 1]

The relationships between impulsivity and mood in bipolar disorder: An ecological momentary assessment study

PONE-D-24-52633R1

Dear Dr. Guzman-Parra,

We’re pleased to inform you that your manuscript has been judged scientifically suitable for publication and will be formally accepted for publication once it meets all outstanding technical requirements.

Kind regards,

Mehdi Rezaei

Academic Editor

PLOS ONE

Additional Editor Comments (optional):

Reviewers' comments:

Reviewer's Responses to Questions

**Comments to the Author**

1. If the authors have adequately addressed your comments raised in a previous round of review and you feel that this manuscript is now acceptable for publication, you may indicate that here to bypass the “Comments to the Author” section, enter your conflict of interest statement in the “Confidential to Editor” section, and submit your "Accept" recommendation.

Reviewer #1: (No Response)

Reviewer #2: All comments have been addressed

2. Is the manuscript technically sound, and do the data support the conclusions?

Reviewer #1: Yes

Reviewer #2: Yes

3. Has the statistical analysis been performed appropriately and rigorously? 

Reviewer #1: Yes

Reviewer #2: Yes

4. Have the authors made all data underlying the findings in their manuscript fully available?

Reviewer #1: Yes

Reviewer #2: Yes

5. Is the manuscript presented in an intelligible fashion and written in standard English?

Reviewer #1: Yes

Reviewer #2: Yes

6. Review Comments to the Author

Reviewer #1: 1. The manuscript has been appropriately and thoroughly revised in accordance with the reviewer’s prior comments. The authors demonstrated close attention to each suggestion and implemented well-justified methodological and editorial improvements.

2. The concern regarding statistical overfitting in a small sample has been properly addressed through a transition to mixed logistic regression models with dichotomized outcomes. The removal of interaction terms and detailed assessment of model assumptions (e.g., heteroscedasticity, multicollinearity) were appropriate and clearly reported.

3. All suggested data visualizations—including a study flowchart, time-series graphs for mood and impulsivity, and a regression plot for lagged effects—have been successfully added and enhance the reader’s understanding of the methodology and results.

4. The Discussion section has been substantially strengthened, particularly through clearer comparisons with previous EMA studies and plausible explanations for divergent findings (e.g., Titone et al., Depp et al.). The inclusion of limitations related to measurement, sample characteristics, and exclusion criteria was also appropriate and transparent.

5. The manuscript now includes a clarified data availability statement, with a direct link to the OSF repository. This ensures full compliance with PLOS ONE’s open data policy.

6. The language is clear, professional, and grammatically correct, and the manuscript is written in standard, intelligible academic English throughout.

7. Overall, the authors have done an excellent job revising the manuscript. Their responses were comprehensive, and the revised version is now a technically sound and well-presented contribution to the literature on bipolar disorder and ecological momentary assessment.

Reviewer #2: 1.Minor Revisions Addressed

1-1.Sample size limitations and lack of power analysis acknowledged.

1-2.Impulsivity/mood measurement limitations clarified.

1-3.Lagged model strategy and decomposition into within-/between-person effects clearly described.

1-4.Clearer definition of “multiplex families” added.

1-5.Consistency and citation issues fixed.

2.Strengths of the Revised Manuscript

2-1.Important and underexplored research question.

2-2.Innovative use of EMA in genetically enriched samples.

2-3.Thoughtful and transparent handling of small sample constraints.

2-4.Open data and open science practices observed.

3.Remaining Minor Suggestions

3-1.Terminology in the Results: “Near to significance” is unclear. Consider replacing with “marginally significant” or simply report the p-values without interpretation.

3-2.Model Justification in Methods:Though addressed in the response letter, a brief explanation in the manuscript text of why dichotomization was necessary (due to assumption violations) would improve transparency for readers.

3-3.Mood Assessment Limitations:Consider briefly discussing how a single-item mood measure may not differentiate between positive and negative affect, which may explain some differences from Titone et al. or Depp et al.

4.Conclusion: The authors have made substantive improvements in response to the first review. The revisions are thoughtful and the manuscript is now methodologically sound, transparent, and scientifically relevant. With minor edits as suggested above, it will be suitable for publication.

7. PLOS authors have the option to publish the peer review history of their article (what does this mean? ). If published, this will include your full peer review and any attached files.

**Do you want your identity to be public for this peer review?** For information about this choice, including consent withdrawal, please see our Privacy Policy .

Reviewer #1: No

Reviewer #2: No

---

## [Editor Report · Acceptance letter]

PONE-D-24-52633R1

PLOS ONE

Dear Dr. Guzman-Parra,

I'm pleased to inform you that your manuscript has been deemed suitable for publication in PLOS ONE. Congratulations! Your manuscript is now being handed over to our production team.

Kind regards,

on behalf of

Dr. Mehdi Rezaei

Academic Editor

PLOS ONE